# Morphological Description of the Early Events during the Invasion of *Acanthamoeba castellanii* Trophozoites in a Murine Model of Skin Irradiated under UV-B Light

**DOI:** 10.3390/pathogens9100794

**Published:** 2020-09-27

**Authors:** Mariana Hernández-Jasso, Dolores Hernández-Martínez, José Guillermo Avila-Acevedo, José del Carmen Benítez-Flores, Isis Amara Gallegos-Hernández, Ana María García-Bores, Adriana Montserrat Espinosa-González, Tomás Ernesto Villamar-Duque, Ismael Castelan-Ramírez, María del Rosario González-Valle, Maritza Omaña-Molina

**Affiliations:** 1Laboratorio de Amibas Anfizoicas, Facultad de Estudios Superiores Iztacala (FESI), Medicina, Universidad Nacional Autónoma de México (UNAM), Tlalnepantla, Estado de México (Edo. Méx) 54090, Mexico; bmaryhj@gmail.com (M.H.-J.); alol_madole@yahoo.com.mx (D.H.-M.); ismaelc.40@gmail.com (I.C.-R.); 2Laboratorio de Fitoquímica, Unidad de Biotecnología y Prototipos (UBIPRO) FESI, UNAM, Tlalnepantla, Edo. Méx 54090, Mexico; tuncomaclovio2000@gmail.com (J.G.A.-A.); boresana@iztacala.unam.mx (A.M.G.-B.); adriana.espinosa@iztacala.unam.mx (A.M.E.-G.); 3Laboratorio de Histología, Unidad de Morfofisiología y Función (UMF), FESI, UNAM, Tlalnepantla, Edo. Méx 54090, Mexico; jdelcjdelc@gmail.com (J.d.C.B.-F.); gvallemr@hotmail.com (M.d.R.G.-V.); 4Facultad de Estudios Superiores Iztacala, Optometría, UNAM, Tlalnepantla, Edo. Méx. 54090, Mexico; amara.gallegos.hernandez@iztacala.unam.mx; 5Bioterio General, FESI, UNAM, Tlalnepantla, Edo. Méx 54090, Mexico; vidutoer@yahoo.com.mx

**Keywords:** *Acanthamoeba castellanii*, murine model, skin invasion, pathogenic mechanism

## Abstract

Skin infections have been associated with *Acanthamoeba*, nevertheless the events during skin invasion and UV-B light effects on it are unknown. The early morphological events of *Acanthamoeba castellanii* skin invasion are shown in SKH-1 mice that were chronically UV-B light irradiated. Mice that developed skin lesions (group 1) were topical and intradermally inoculated with *A. castellanii* trophozoites and sacrificed 48 h or 18 days later. Mice that showed no skin lesions (group 2) were intradermally inoculated and sacrificed 24, 48 or 72 h later. Mice ventral areas were considered controls with and without trophozoites intradermally inoculated. Skin samples were processed by histological and immunohistochemistry techniques. In group 1, trophozoites were immunolocalized in dermal areas, hair cysts, sebaceous glands, and blood vessels, and collagen degradation was observed. One of these mice shown trophozoites in the spleen, liver, and brain. In group 2, few trophozoites nearby collagenolytic activity zones were observed. In control samples, nor histological damage and no trophozoites were observed. Adherence and collagenolytic activity by *A. castellanii* were corroborated in vitro. We can infer that UV-B light irradiated skin could favor *A. castellanii* invasiveness causing damage in sites as far away as the brain, confirming the invasive capacity and pathogenic potential of these amphizoic amoebae.

## 1. Introduction

The general amoebae biological characteristics include ubiquitous distribution, use of soil and water habitats, air use as dispersal media, and ecologically they participate in bacterial population control [1,2]. However, *Acanthamoeba* spp. are etiology agents of human pathologies, such as amoebic keratitis (AK), granulomatous amoebic encephalitis (GAE) and skin infections [2]. These protozoans have been more appropriately designated as amphizoic microorganisms, due to their ability to exist in the environment as free-living organisms or as opportunistic parasites. Until now, there are no accurate epidemiological records related to the incidence of pathologies caused by these amoebae [3]. Diagnosis is usually carried out long delayed, after weeks or months of infection, and currently there is no treatment for affected patients that is 100% effective [4]. The GAE is a subacute or chronic infection causing necrosis and inflammatory lesions, whose entry route can be hematogenous or through the respiratory tract, affecting mainly immunocompromised people [5], with multiple sclerosis or diabetes [2,6,7]. Nevertheless, clinical GAE cases have occurred in immunocompetent people and patients with organ transplants [8,9,10,11]. No less important is the AK—a painful corneal infection with a chronic course that usually occurs in contact lens wearers, characterized by, progressive corneal epithelium necrosis and stromal lamella destruction, which can cause a decrease in visual acuity, the loss of vision or eyeball [12,13]. On the skin, amoebae proliferate and induce ulcers in people who have suffered skin lesions. Risk factors for *Acanthamoeba* skin infection include traumatized areas, such as surgical scars, skin lesions caused by the varicella-zoster virus, bites [14], mechanical trauma [15] and burns [16,17,18]. It is widely accepted that skin infections occur mainly in people with a weakened immune system, with AIDS, although it does not always affect the central nervous system (CNS) [19,20,21]. In addition, skin infections have also been reported in HIV-negative GAE patients, as well as in patients who are undergoing immunosuppressive therapy after organ transplantation or with immunological disorders [3,22,23]. Similarly, *Acanthamoeba* skin infections may be the manifestation of an infection that initiated in another organ and spread hematogenously or are the primary focus of infection through skin wounds or burns and that posteriorly spread to other organs [21,24,25].

Another species of amphizoic amoebae related to skin infections is *Balamuthia mandrillaris.* It has been suggested that environmental factors such as UV-B light predispose to infection caused by these amoebae, as well as related activities to gardening/soil [26]. These risk factors have not been evaluated in *Acanthamoeba.*

Currently, the early events that occur during the invasion of amoebae of *Acanthamoeba* genus into the skin are unknown, as well as if environmental factors, such as UV light, predisposes the invasion of amoebae into the skin.

The present work focuses on the study of the morphological events carried out by *A. castellanii* trophozoites during the invasion of skin from SKH-1 mice previously exposed to chronic irradiation with UV-B light. In addition, the ability of amoebae to migrate from a primary skin focus to the CNS and various organs was also evaluated.

## 2. Results

### 2.1. Murine Skin Infection

Out of the five SKH-1 mice chronically irradiated with UV-B light, in two of them (group 1) it was confirmed evident damage caused by UV-B light chronic irradiation (Figure 1A,B). In group 2 mice showed no evident lesions after chronic UV-B irradiation (Figure 1C).

### 2.2. Histological and Immunohistochemistry Analysis of the Interaction of A. castellanii Trophozoites with Skin Chronically Irradiated with UV-B Light

Group 1: The mouse that developed an erythematous lesion was sacrificed 48 h post-interaction as mentioned in the methodology. As expected, in both control ventral zones (inoculated with saline solution or with *A. castellanii* trophozoites) no evident histologic changes and no trophozoites were observed (Figure 2A). In the histological and immunohistochemical analysis of the damaged area in which trophozoites were placed topically, numerous amoebae were observed in trophic form in optimal conditions along the dermis, attached to adipocytes and adhered to the collagen fibers and interfibrillar spaces (Figure 2B–D). In addition, trophozoites adhered to connective tissue, near blood vessels and hair cysts were observed. Collagenolytic activity was detected near areas with trophozoites (Figure 2E–F).

Likewise, in histological sections of the intradermally inoculated areas, numerous trophozoites were observed invading the dermis and migrating to the hypodermis. Besides, amoebae were located attached to adipocytes and adhered to connective tissue, as well as surrounding and near blood vessels in both hypodermis and muscle (Figure 3A–E). Control ventral zone did not show evident histologic changes or the presence of trophozoites (Figure 3F).

As mentioned in the methodology, the mouse presenting neoplasic lesion (squamous cell carcinoma) (1.5 cm) was maintained under observation for several days, in order to evaluate possible invasion of the amoebae from the zone of interaction in the skin into other organs. It occurred 18 days post-interaction, presenting clinical signs such as lethargy and tremors. For this reason, it was decided to sacrifice the mouse. Analysis of the lesion (Figure 1B) showed neoplasia, with the histological characteristics of squamous cell carcinoma, with areas of irregular epithelial thickening due to hyperplasia and hyperkeratosis, keratin beads and epidermal invasion into the dermis and absent basal membrane (Figure 4A). In this area of the lesion, where trophozoites were placed on topically, amoebae were observed alone or in small groups located in the epidermis and dermis; inside hair cysts, associated with adipocytes, and in the wall of a blood vessel; collagen degradation is also observed in areas close to trophozoites (Figure 4B–E). Regarding the samples inoculated intradermally at the different point times, the results are very similar to those observed in the mouse that developed erythematous lesions. As expected, in the tissue of control areas, well–organized and differentiated skin layers were observed, with the hair cysts characteristic of the SKH-1 mouse strain at the hypodermis (Figure 4F).

Histological sections of the spleen, liver and brain of the mouse sacrificed 18 days post-interaction were analyzed, finding trophozoites but no cyst (Figure 5A–C). It is important to highlight that in the spleen we observed hypertrophy and hyperplasia. Abundant trophozoites were observed in the white pulp-forming cells of the spleen (Figure 5A). In the liver, trophozoites were observed adhered to hepatocytes (Figure 5B). At brain level in some areas, we observed a moderate glial reaction in which trophozoites were immunolocalized (Figure 5C).

Group 2: Mice without apparent lesions sacrificed 24, 48 and 72 h post intradermal inoculation. Trophozoites were observed in optimal conditions and constantly along the dermis attached between the collagen fibers and interacting with dermal components such as sebaceous. No differences were observed between the evaluated times (Figure 6A–C).

Control group (trophozoites inoculated in ventral zones). Ventral areas showed well-organized collagen fibers at the dermis. At hypodermis, the mice strain characteristic hair cysts were observed. Histologically normal tissue was observed: well-organized layers of skin and without the presence of amoebae (Figure 6D).

### 2.3. In Vitro Determination of the Adhesion of A. castellanii Trophozoites to Type I Collagen and Its Collagenolytic Activity

Trophozoites adhered strongly to type I collagen; since the first 5 min, 15% of trophozoites adhered to the collagen film and about 80% of the amoebic population did so within 30 min of the interaction.

Collagenolytic activity was expressed as the percentage of degradation of a type I collagen film after 16 h of interaction. The trophozoites of the strain under study showed moderate activity, degrading 40% of the substrate, the conditioned medium showed significantly less activity by degrading only 4% of the collagen substrate.

## 3. Discussion

The implementation of an experimental model of cutaneous infection by amoebae of the genus *Acanthamoeba* is necessary to understand the mechanisms involved in the pathogenicity of these amphizoic amoebae in different target tissues, which possibly contributes to the compression of physiological processes that participate during the invasion of amoebae, as well as the search for better diagnoses and therapies, which allow an earlier resolution of the infection if possible, minimizing the damage that amoebae can induce to the skin and other organs. Since the pathogenic potential of amphizoic amoebae of *Acanthamoeba* genus was demonstrated, several assays have been carried out on different animal models, with the purpose of describing the histopathological findings of experimental infections induced by these amoebae in the cornea and mainly in the CNS. These findings were reported in healthy as well as diabetic organisms, in periods of time ranging from 3 to 14 days, and even up to 7 months post-inoculation [5,13,27,28,29,30].

Until now, the early invasion events that take place since *Acanthamoeba* spp. comes into contact with the skin until amoebae induce injury had not been reported.

Based on the results obtained in this study, we consider that through the in vivo experimental model in which the SKH-1 mouse was used, it was possible to demonstrate that the trophozoites of *A. castellanii*, a common species of AK and GAE cases in humans, are capable of invading and causing damage to chronically irradiated skin—being the main focus from which they invade different organs, including the CNS.

During the implementation of the murine model of skin irradiated with UV-B light, UV-B radiation was found to cause skin lesions in SKH-1 mice; an erythematous lesion and a lesion related to the development of neoplasic lesion (squamous cell carcinoma) which coincides with Mäkinen and Stenbäck [31] and Cano et al. [32], who reported that excessive exposure to UV-B is carcinogenic and that epidermal and dermal lesions are easier to be induced by UV-B radiation in SKH-1 mice. In addition, a great similarity has been recognized between the lesions developed in experimental animals with those that take place in the human skin after chronic exposure to ultraviolet radiation [33]. For that reason, the nude SKH-1 mice established model was useful for the study of *A. castellanii* infection in damaged or undamaged skin.

The *A. castellanii* trophozoites invasion process begins with the adhesion of amoebae to the surface of the epithelium where they were placed, or from where they were inoculated. Trophozoites were able to penetrate the epithelial layers, subsequently reach the dermis, hypodermis, and muscle, showing avidity for hair cysts. It is relevant to highlight the erythematous lesion in which an integral epithelium is observed, which means that amoebae were able to penetrate it, since, as is known, erythema is an inflammatory process that occurs in the dermis and does not significantly involve the epithelium.

The histopathological and immunohistochemical findings during the interaction of amoebae in damaged and undamaged skin allowed us to suggest that contact-dependent mechanisms, such as adhesion, migration to deeper layers of the skin and penetration of the amoebae are relevant pathogenic mechanisms [28,34]. This is supported by the fact that through the results shown in this study, the amoebae invaded the different layers of the skin; epidermis, dermis and hypodermis and, in particular, trophozoites adhered to hair cysts, adipocytes and blood vessels, and even invaded the muscle layer.

In the skin lesions, the squamous cell carcinoma (18 days post-interaction) and the erythematous lesion (48 h post-interaction), where the amoebae were placed topically, a greater amount of viable trophozoites were observed in optimal conditions and constantly in all skin layers: epidermis, dermis and hypodermis, as well as in muscle, interacting with dermal components such as hair cysts and adipocytes. Particularly *A. castellanii* trophozoites were constantly located along the dermis, mainly around blood vessels and in the subcutaneous fatty tissue, which is consistent with histopathological studies of skin lesions of patients infected with *Acanthamoeba* [35]. Cyst forms were not observed in any of the samples evaluated, reinforcing the idea that amoebae were found in an optimal environment for their development and invasion.

Apparently, as previously described by Omaña-Molina et al. [28], amoebae alone or in groups are capable of invading and causing damage. In our study, we observed a few amoebae causing damage in nearby areas where they were found. It is important to highlight the low inflammatory response of the host tissues, which is in agreement with other studies performed in vivo in the CNS, in which the amoebae in short interaction times (24–96 h) did not induce a relevant inflammatory response [29,30].

The immunohistochemical technique allowed us to demonstrate the presence of trophozoites of *A. castellanii*, since it was possible to observe their classical morphology, their integrity and location. Through this assay, we observed the amoebae in situ on the target tissue that they invaded—the skin, lung, spleen, brain, where it is possible to observe histopathological characteristics such as inflammation, collagenolytic activity and blood vessels invasion. In addition, the antibody directed against the amoeba and the technique itself makes it possible to reveal the trophic forms, which was corroborated with the controls, so it was not necessary to implement other tests for their detection.

Although through the histological sections and immunohistochemical techniques we demonstrated the invasion of the trophozoites of *A. castellanii* in the skin, it was not possible to show the process in greater detail, however, we propose that these protozoa carry out mechanisms similar to those described by transmission and scanning electron microscopy in previous studies with the same strain interacting with other target tissues such as the cornea and the CNS [28,29,30,36]. It is possible to suggest a sequence of events that start with the adhesion of the trophozoites [37], migration towards the cell junctions of the target cells and invasion through a paracellular route, playing an important role in the detached cells phagocytic processes, which together alter tissue architecture. It is possible that amoebae, during their invasion, degrade and feed on the extracellular matrix, which was corroborated evaluating the adherence and degradation of type I collagen in vitro. In addition, they can also feed on sebaceous glands and the content of hair cysts.

The amoebae of the genus *Acanthamoeba* are aerobic organism, therefore the presence of trophozoites surrounding blood vessels suggests that amoebae migrate to areas with higher oxygenation which also allow them to reach the circulatory system for spreading to other organs (reaching the CNS via the hematogenous route). *A. castellanii* trophozoite invasion was corroborated 18 days post amoebae interaction on skin carcinoma, because the mouse presented clinical signs such as lethargy and asterixis (tremor). This was verified by analyzing histological samples of the brain and observing trophozoites in it, which suggested a hematogenous invasion through the blood vessels to the brain; likewise, trophozoites were found in spleen and liver, which indicates a spread to the rest of the organs. *A. castellanii* spread hematogenously relatively fast since they were able to reach the organs within a few days after inoculation. Once in organs, the amoebae must invade them, which is carried out more slowly with respect to that described with *Naegleria fowleri*, a highly virulent amoeba, which would explain the chronic and insidious course of infections induced by *Acanthamoeba* spp., highlighting the lack of an evident inflammatory host response. The host immune response to *Acanthamoeba* spp. infection is poorly understood, however, in the particular case of the invasion of amoebae to the spleen, this organ showed hyperplasia and hypertrophy and the presence of numerous amoebae with respect to the other organs analyzed, which would possibly indicate a response from the host. Łanocha-Arendarczyk et al. [38] reported that *Acanthamoeba* spp. induced a change in spleen weight during intraperitoneal inoculation. Initial amoebae adherence to the target cell may be mediated through specific receptors [39,40]. Cao et al. [41] demonstrated that adhesion mediated by mannose-binding proteins contributes to the induction of a cytopathic effect given by the parasite, which probably favors penetration and invasion of the skin layers until reaching the muscle and eventually penetrating capillaries and vessels to continue invasion to other organs. We consider that the events that take place are a combination of dependent and independent contact events, where mechanical and enzymatic processes participate. As mentioned before, the analysis of the histological sections and the immunolocalization of the amoebae suggested that *A. castellanii* efficiently adhered to type I collagen and showed collagenolytic activity, which was corroborated in vitro. It is worth mentioning that collagenolytic activity is observed near areas with trophozoites, which is consistent with He et al. [42] and Rocha-Azevedo et al. [43] who reported collagenase activity in axenic cultures of *Acanthamoeba*, as well as lower concentrations of other proteolytic enzymes. In addition, infiltrated and activated neutrophils in turn can exacerbate collagen lysis and keratocyte necrosis through the release of several nonspecific lysosomal enzymes and oxygen metabolites. It has been suggested that the stromal necrosis characteristic of amoebic keratitis is the result of collagenolytic enzymes released from neutrophil lysosomes that infiltrate infected corneas [44]. Moreover, diffuse areas of collagen have been reported in skin samples from patients infected with *Acanthamoeba*, suggesting a collagenolytic activity by the amoebae [45]. It is probable that enzymes favor the lysis of the extracellular matrix and so facilitate the invasion of the tissues by the amoeba [46], likewise, in addition to breaking of the extracellular matrix it could phagocyte it and use it as a food source. Similarly, the way *A. castellanii* invades the cutaneous tissue resembles the way other pathogenic amoebae do it, such as *Entamoeba histolytica* [47] and *Balamuthia mandrillaris* [26]. The absence of amoebae in the control areas shows that the damage generated by chronic exposure to UV-B radiation allows the invasion of *A. castellanii* trophozoites in the skin.

The manner in which *A. castellanii* invaded damaged and undamaged skin also suggests analogous pathogenic mechanisms with other pathogenic species of the genus. For that reason, it would be convenient to evaluate different *Acanthamoeba* species under the same experimental conditions in order to generalize that the sequence of events described corresponds to the mechanism’s pathogenicity effects of amoebae of the genus *Acanthamoeba* on the skin.

## 4. Conclusions

Our results suggest that it is not necessary for a large number of trophozoites to reach the blood vessels to initiate hematogenous spread, which gives the possibility of a successful invasion of *A. castellanii,* confirming the invasive capacity and pathogenic potential of these amphizoic amoebae.

With the aforementioned, it can be concluded that through the murine model of skin damaged by irradiation with UV-B light, it was shown that *A. castellanii* trophozoites are capable of invading areas of skin exposed or damaged by this type of radiation, in addition to invading organs such as the liver, spleen and brain tissue, hematogenously with a primary focus on skin.

## 5. Material and Methods

### 5.1. Mice Maintenance and Experimental Use, Ethical Considerations

Animal care and use protocols were in accordance with the Official Mexican Standard NOM-062- ZOO-1999. Animals were kept at the UNAM-FES Iztacala Bioterio General, in microisolator systems, and in controlled environmental conditions: temperature (23–25 °C), relative humidity (45–55%) and light–dark cycle (12:12 h). Microisolator included substrate (Sanichips, ENVIGO, Mexico City, Mexico), at libitum water and food (ENVIGO 2018s), and enough space in accordance with mice size. All materials were changed twice a week by recently sterilized ones. Biosecurity Class II Type A2 cabinet (Nuaire, MN, USA) was used for animal manipulation. The project was carried out with the approval of the UNAM FES Iztacala Ethical Committee: CE/FESI/092,017/1199.

### 5.2. Amoebic Cultivation

The strain of *A. castellanii* (T4 genotype) was isolated from a clinical case of AK from “La asociación para evitar la ceguera en México”, Luis Sánchez Bulnes Hospital, Mexico City. It was previously demonstrated that this strain is invasive in the GAE murine model [30,48].

The amoebae growth and maintenance were under axenic conditions, in PYG medium with added antibiotics (10,000 U/mL penicillin and 10 mg/mL streptomycin), and incubated at 30 °C. After 72 h of incubation (end of growth logarithmic phase), culture was chilled (4 °C), centrifuged (2500 rpm during 5 min) and trophozoites were harvested for the subsequent assays.

### 5.3. Reactivation of A. castellanii Virulence

Even though the strain in the study was considered pathogenic since it was isolated from an AK case, its prolonged maintenance in axenic culture could imply a diminished virulence. Virulence reactivation was performed by trophozoites intranasal administration in three-week-old male BALB/c mice. Three serial passes of trophozoites were carried out by intranasal administration. Briefly, trophozoites pellet was diluted with fresh culture medium (without antibiotic), obtaining 1 × 10^6^ trophozoites in 20 μL for each inoculum. The viability of the trophozoites was determined by trypan blue (0.4%) technique. Three mice were anesthetized with isoflurane vapors and inoculated into each nostril, to promote CNS infection in accordance with Culbertson et al. [49]. Twenty-one days post infection, the remaining living mice were sacrificed with isoflurane vapors. For amoebae recovery, the brain, lungs, liver and kidney were deposited on non-nutrient agar with added heat-inactivated *Enterobacter aerogenes* (NNA). Finally, we axenized trophozoites in PYG medium before subsequent experiments.

Two of the five BALB/c mice inoculated intranasally with *A. castellanii* trophozoites died before 21 days (7 days post-inoculation), time established by Culbertson et al. [49] for this procedure, while the remaining mice were sacrificed once the 21 days elapsed. The brain, lungs, liver and kidneys were extirpated. A macroscopic analysis was performed in which no morphological or pathological changes such as edema, necrosis or hemorrhage were observed. After two days of incubation, through a light microscope, it was observed that trophozoites emerged from all the organs, confirming virulence and invasive capacity of the strain in study. Amoebae recovered from the brain were axenized again in the PYG medium and used in the proposed assays.

All the assays took place with a trophic culture, since the conditions in which the amoebae were grown ensured that we worked with a homogeneous population of trophozoites; in addition, having worked with amoebae recently recovered from the mouse brain ensured that this strain had the optimal virulence to carry out the assays proposed.

### 5.4. SKH-1 Mice Chronic Irradiation with UV-B Light

Five female SKH-1 mice, 6 to 8 weeks old (weighing 26 ± 5 g) were purchased from Charles River Laboratories (Wilmington, MA, USA) and were acclimatized at least one week before performing the experiments.

At the time of irradiation, the experimental animals were placed in special acrylic boxes, 15 cm away from the UV-B light lamp (312 nm, Spectroline EB-280C) (Spectronics Corporation, Westbury, NY, USA). The irradiation energy at this distance is 6.0 mJ/cm^2^ (field strength 130 watts/m^2^) calculated with a radiometer (Spectroline DM-300HA); the lamps were systematically alternated to compensate or minimize changes in their flow. A chronic irradiation scheme consisted of exposure to UV-B light for 1 min each day for two weeks. Subsequently, mice were irradiated three times a week for 1 min for 32 weeks [50].

### 5.5. Murine Model of A. castellanii Skin Infection

At the end of the chronic UV-B light irradiation period, two of the mice had developed macroscopic skin lesions, consequently, it was decided to work according to the following schemes of interaction with the amoebae:

Group 1: Two mice showed macroscopic skin lesions: one of them developed a 5 mm erythematous lesion and the second one developed an apparent squamous cell carcinoma (1.5 cm). In both, the interaction of amoebae with skin lesion was carried out by placing 2.5 × 10^4^/20 μL trophozoites of *A. castellanii* directly on the lesion area, besides, in a parallel assay, amoebae were inoculated intradermally in five pre–established areas of the back in each mouse using ultrafine needles. Due to no experimental antecedents related to the invasion of *Acanthamoeba* in the skin, we decided to sacrifice, 48 h post-inoculation, the mouse who developed an erythematous lesion. For the second mouse who developed squamous cell carcinoma, after performing the interaction with the amoebae, we decided to wait for the time required to observe signs of CNS infection, which occurred 18 days post-interaction. Both mice were sacrificed with isoflurane vapors.

Group 2: The remaining three mice, with no evident skin lesions (only erythema was observed), were inoculated intradermally with 2.5 × 10^4^/20 μL trophozoites in 5 pre–selected dorsal areas, and were euthanized at 24, 48 and 72 h later.

Control groups: The ventral area of the mice was considered as a control, since it was not a UV-B light irradiated area. Two kinds of control skin areas were considered: a negative control with no *A. castellanii* inoculation (inoculated with saline solution) (C0) and a positive control with intradermal *A. castellanii* inoculation (C1). In the C1 group, five ventral points in each mouse were selected and inoculated with the same amount of trophozoites (2.5 × 10^4^/20 μL). At the end of each time of interaction, the animals were sacrificed with isoflurane vapors. In all assays (including control samples), 1 cm^2^ sections of skin were obtained, considering the inoculation site or lesion area as the center of them.

Finally, skin samples were fixed with 4% paraformaldehyde and processed by conventional histological technique, which includes dehydration, inclusion in paraffin and sectioning to 3–5 μm, which were stained with H&E technique [51].

### 5.6. Immunohistochemistry Analysis of the Interaction between A. castellanii Trophozoites and Skin from SKH-1 Mice

Histological samples were processed by paraffin removal, rehydration and washing with TBS-Tween 20 (0.1%) (TBS-T). For antigenic recovery, we digested enzymatically with K proteinase 0.1 mg/mL in TBS-T buffer with 1% CaCl_2_ during 15 min. Endogenous peroxidase was blocked with 3% H_2_O_2_ for 15 min. Slides were washed, incubated with 5% fetal bovine serum during 2 h, and incubated during the night with rabbit polyclonal antibodies anti-*A. castellanii* (1:100). We used 5 heat shock cycles to promote lysis of the trophozoites (95%) and cysts (5%) mixture to obtain the polyclonal antibody and a conventional immunization scheme was followed in an adult male New Zealand rabbit. Afterward, we used TBS-T for samples washing, and incubated them with the secondary antibody (HRP-Rabbit MACH 2 Polymer Biocare Medical, CA, USA), for 30 min. Samples were treated with diaminobenzidine–H_2_O_2_ (DAB Peroxidase Substrate, Vector Laboratories Inc., Burlingame, CA, USA) and counter-stained with Harris hematoxylin to evaluate peroxidase activity. Samples were dehydrated and covered with synthetic resin. Procedures for negative controls were the same, but without primary antibodies 

The slides were microscopically observed (Nikon, Eclipse E400, Tokyo, Japan), evaluating trophozoites or cysts presence, histopathological changes, as well as inflammatory cells with the purpose of making a photographic record.

### 5.7. Adhesion and Collagenolytic Activity of A. castellanii Trophozoites to Type I Collagen

Type I collagen of human origin was extracted from placentas according to the technique described by Muñoz et al. [52].

Adherence assay: The adherence to collagen type I of *A. castellanii* trophozoites was determined at 5 and 30 min. A 40 μL film of type I collagen (3 mg/mL) was placed on 96-well cell culture plates and polymerized with UV light for 1 hr. Subsequently, 5 × 10^4^/200 μL trophozoites were placed in PYG medium in each well and then incubated at 37 °C at different point times. Then, washed with PBS at 37 °C to remove unadhered trophozoites; finally, cold PBS (4 °C) was added to remove those that did adhere and count them in a Neubauer chamber.

Collagenolytic activity of the trophozoites and their conditioned medium was determined according to the technique described by Muñoz et al. [52]. Trophozoites in PYG medium (5 × 10^4^/200 μL) were placed in 96-well plates with type I collagen as described above, and then incubated during 16 h at 37 °C. Posteriorly, samples were washed with a PBS solution at 4 °C. EDTA (25 mM) was added for one hour to inhibit the adherence of the trophozoites to the substrate. Afterward, the plate was washed several times with PBS. The remaining collagen film was fixed with 4% formaldehyde and stained with a saturated solution of Sirius red in picric acid for 1 h, the excess was washed with a 0.001 N HCl solution and was eluted with 0.1 N NaOH in absolute methanol (1:1). Samples were read on a spectrophotometer at 540 nm wavelength. The percentage of collagen degraded by amoebae was determined. Based on the spectrophotometric reading, the collagen eluted in the control wells (without amoebae or conditioned medium) was considered 100% of the collagen concentration in each well and was used as a reference value to calculate the percentage of degraded collagen in the experimental wells from the remaining collagen.

The same procedure was carried out using conditioned medium, which is the culture medium in which the amoebae were cultivated. Obtaining the conditioning medium is briefly described: 5 × 10^6^
*A. castellanii* trophozoites were cultivated in 5 mL of medium without serum for 24 h, then trophozoites were detached with ice water (4 °C) and centrifuged for 5 min at 3000 rpm. The supernatant was recovered and filtered (0.22 microns). Both assays were carried out in triplicate.

## Figures and Tables

**Figure 1 pathogens-09-00794-f001:**
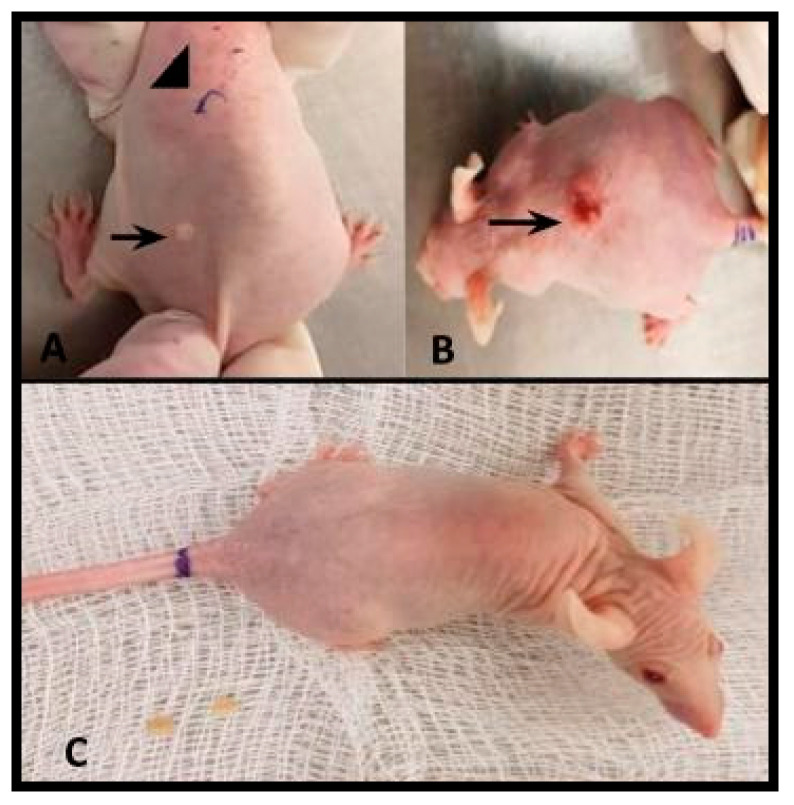
(**A**,**B**) Group 1 SKH-1 strain mice, with skin lesions caused by chronic UV-B light irradiation. (**A**) Erythematous lesion (arrow) where trophozoites were placed on topically and skin without apparent lesion (arrowhead) in which intradermal inoculation of trophozoites was performed. (**B**) Ulcerated neoplasia (squamous cell carcinoma) of approximately 1.5 cm diameter in which trophozoites were placed on topically. (**C**) Group 2 SKH-1 mouse without skin lesions after being irradiated with UV-B light and intradermally inoculated with *A. castellanii* trophozoites.

**Figure 2 pathogens-09-00794-f002:**
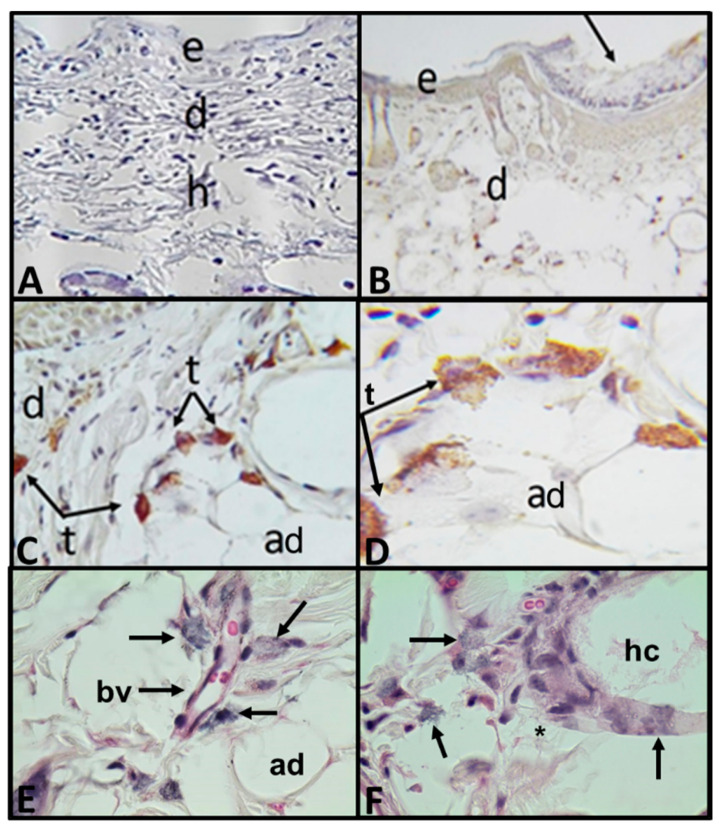
Photomicrographs of histologic skin sections with erythematous lesion caused by UV-B light in SKH-1 mouse sacrificed 48 h post-topical interaction with *A. castellanii* trophozoites. (**A**) Control, ventral zone, without evident histologic changes or trophozoites (×100). (**B**–**D**) Immunolocalization of trophozoites from *A. castellanii.* (**B**) Panoramic view of the injured area (arrow) where the amoebae were placed topically (×100). (**C**) Approach of the lesion area with trophozoites near dermis and adipocytes (×400). (**D**) Numerous trophozoites in the dermis associated with adipocytes (×1000). (**E**) Trophozoites (arrows) in a blood vessel wall in dermal connective tissue. Adipocytes are observed at its side (×1000). (**F**) Trophozoites (arrows) in the wall of a hair cyst where collagen lysis is observed (*) (×1000). H&E stain. Epidermis (e), dermis (d), adipocytes (ad), hypodermis (h), trophozoites (t), blood vessels (bv) and hair cyst (hc).

**Figure 3 pathogens-09-00794-f003:**
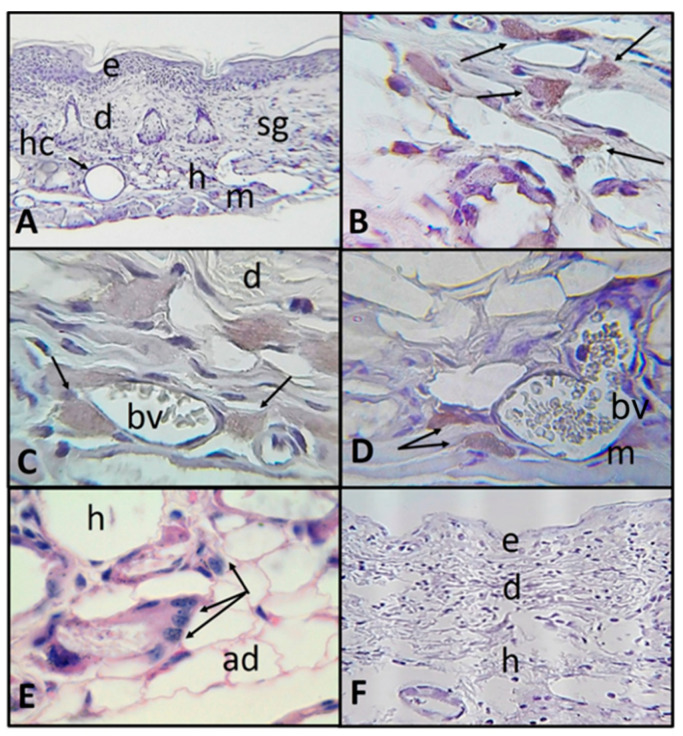
Photomicrographs of histologic skin sections damaged with UV-B light in SKH-1 mice sacrificed 48 h post-intradermal inoculation with *A. castellanii*. (**A**–**D**) Immunolocalization of *A. castellanii* trophozoites (**A**) panoramic view (×100). (**B**) Trophozoites (arrows) in dermis and hypodermis, close to adipocytes (×400). (**C**) Trophozoites in dermis and hypodermis, surrounding a blood vessel (×1000). (**D**) Trophozoites (arrows) in muscle, near blood vessels (×400). (**E**) Trophozoites (arrows) in hypodermis near adipocytes (×1000) (H&E). (**F**) Control, ventral area without evident histologic changes or trophozoite labeling (×100). Epidermis (e), dermis (d), hypodermis (h), muscle (m), sebaceous glands (sg), adipocytes (ad), blood vessel (bv) and hair cyst (hc).

**Figure 4 pathogens-09-00794-f004:**
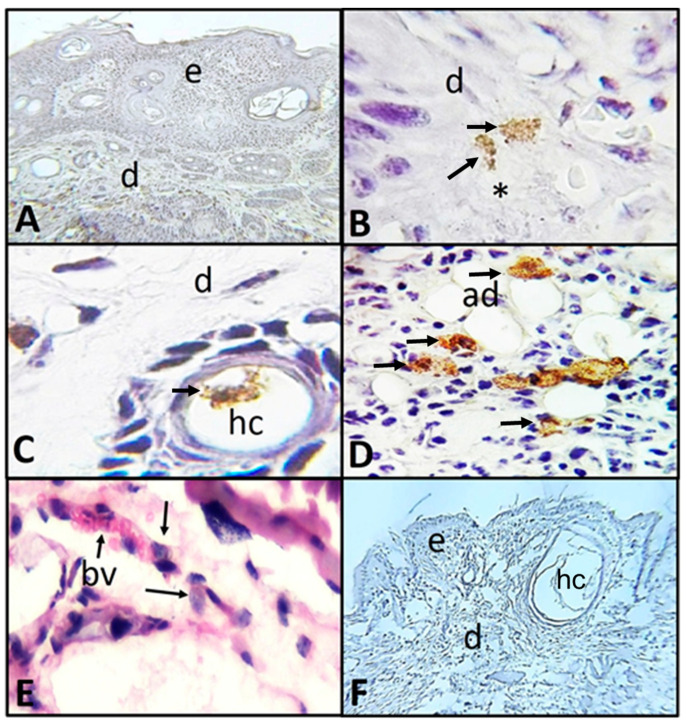
Photomicrographs of histologic skin sections with carcinoma caused by UV-B light in SKH-1 mice sacrificed 18 days post-interaction with *A. castellanii*. (**A**–**D**) Immunolocalization of trophozoites from *A. castellanii.* (**A**) Panoramic view, the neoplasic epidermis and the dermis are identified with trophozoites (×100). (**B**) Trophozoites (arrows) in the dermis collagenolytic activity (*) were observed near areas with trophozoites (×1000). (**C**) Amoebae (arrow) inside a hair cyst (×1000). (**D**) Trophozoites (arrows) associated with adipocytes (×1000). (**E**) Trophozoites (arrows) are observed in connective tissue and in the wall of a blood vessel (H&E). (**F**) Control ventral zone without evident histologic changes or trophozoites (×400). Epidermis (e), dermis (d), hair cyst (hc), adipocytes (ad), connective tissue (ct) and blood vessel (bv).

**Figure 5 pathogens-09-00794-f005:**
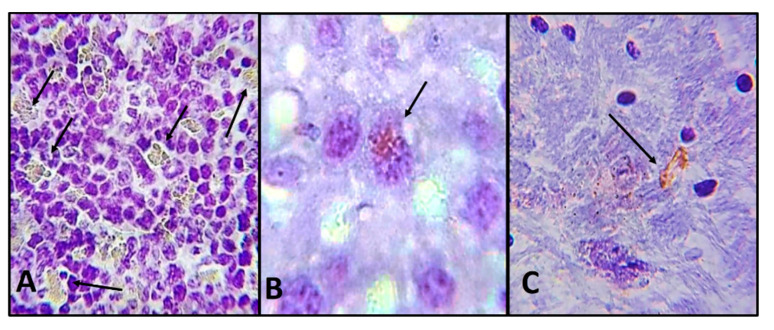
Photomicrographs of histologic sections of SKH-1 mouse organs which developed neoplasic lesion in the skin caused by UV-B light and sacrificed 18 days post *Acanthamoeba* interaction. (**A**) Spleen. Abundant trophozoites (arrows) are observed in the white pulp-forming cells of the spleen (×400). (**B**) Liver. Trophozoite (arrow) adhered to hepatocyte (×1000). (**C**) Brain. Trophozoite (arrow) located in nervous tissue (×1000). Immunolocalization.

**Figure 6 pathogens-09-00794-f006:**
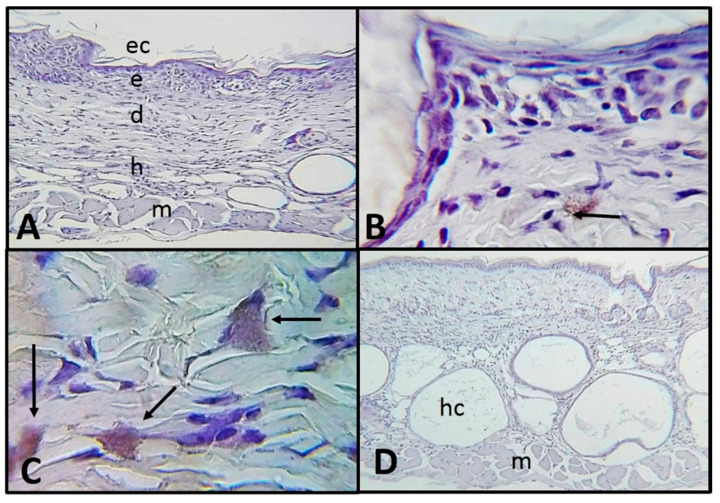
Photomicrographs of histologic sections of SKH-1 mouse skin exposed to UV-B light and sacrificed 24, 48 and 72 h post-intradermal inoculation with *A. castellanii*. Results were remarkably similar in all times evaluated. (**A**) Panoramic view (×100). (**B**) Trophozoite (arrow) in dermis (×400). (**C**) Trophozoites (arrows) in dermis (×1000). (**D**) Control, ventral zone without evident pathological changes and without trophozoite labeling; the typical hair cysts of the mouse strain are observed (×100). Immunolocalization. Stratum corneum (ec), epidermis (e), dermis (d), hypodermis (h) muscle (m) and hair cyst (hc).

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
