# Peer review of "Morphological Description of the Early Events during the Invasion of Acanthamoeba castellanii Trophozoites in a Murine Model of Skin Irradiated under UV-B Light"

_pathogens, 2020, doi:10.3390/pathogens9100794_

Round 1

Reviewer 1 Report

The manuscript entitled ‘’Morphological Description of the Early Events During the Invasion of Acanthamoeba castellanii Trophozoites in a Murine Model of Skin Irradiated Under UV-B Light’’ presented a study to investigate on Acanthamoeba castellanii trophozoites invading areas of skin exposed or damaged by UV-B Light. The authors try to investigate the events that during the invasion of amoebae into the skin and the UV-B light effects. The study has the potential to be informative but more details and data need to be shown in the paper. The study is simplistic and offers no great insights into the relationship. The design in this study is too limited.

Reviewer 2 Report

In the manuscript entitled “Morphological description of the early events during the invasion of Acanthamoeba castellanii trophozoites in a murine model of skin irradiated under UV-B light” it is described that A. castellanii is capable of invading skin affected by UV-B light and migrate to other organs.

The following are some comments and suggestions that need major review:

As a general comment, although the article can be generally understood, grammar English mistakes can be found, which make it difficult to understand certain passages of the text. Therefore, I strongly recommend that English is professionally revised so that the manuscript is fully understood and publishable.

ABSTRACT:

Lines 28 and 29: add “which” after “Mice”

Line 37: change Acanthamoeba castellanii for A. castellanii. It has been already mentioned in the abstract.

Line 37: “are” should be changed for “were” to be consistent with the tense used.

KEYWORDS:

The keyword “murine model of Acanthamoeba skin invasion” is too long

INTRODUCTION:

Line 44: the word “cosmopolitan” is not commonly used for FLA. A better word is “ubiquitous”

Lines 45-46: “playing an important…..of nutrients” this affirmation cannot be found in reference 1

Lines 47-48:  central ervous system is not a paghology

Line 51: Provide a reference after “these amoebae”

Line 56: persons vs people. Be consistent along the text. I would suggest to use people

Line 58: Reformulate “a painful difficult to resolve corneal infection”. It is not understandable

Line 81: add “also” before “evaluated”

RESULTS:

Line 89: A. castellanii should be written in italics. Revise this along the text

Line 90: Provide a reference for Culbertson procedure

Lines 94-95: provide the full name for E. aerogenes. It should be in italics.

Line 104: Add the word “Out” at the beginning of the sentence

Line 118: “place” should be changed for “placed”

Lines 147-148: this sentence lacks the verb and cannot beunderstood

Line 161: Add full stop after “tremors”

DISCUSSION:

Lines 225-226: This sentence contains a mix of 2 ideas. It cannot be understood the way it is written

Line 233: add a full stop after the references

Line 234: a reference should be provided

Line 237: “in vivo” should be written in italics. Revise along the text

Line 250: change Acanthamoeba castellanii for A. castellanii. Revise along the text

Lines 261-263: Revise this sentence. It cannot be understood.

Line 274: Add a full stop after “damage”

Lines 279-281: Revise this sentence. It cannot be understood.

Line 286: “in vitro” should be written in italics. Revise along the text

Line 292: Add a full stop after “(tremor)”

Line 296: change “are” for “were” to be consistent. Revise this along the text

Line 330: Add a full stop after “genus”

MATERIALS AND METHODS:

Line 350: specify that T4 is a genotype

Line 351: The very same isolate or the same genotype?

Line 361: hoy many BALB/c mice?

Line 363: 20ul/each inoculum?

Line 378: Do you mean UV-B light?

Line 390: specify how you sacrifice them as you do below

Line 397: why not using non irradiated mice as control group?

Line 407: provide a reference

Reviewer 3 Report

General comments

The study is original, valuable and the obtained results gain interest. Included photos are of the highest quality and value which is advantage of this manuscript. Results are clear and well presented, however manuscript could be updated with the discussion regarding some questions that came out after reading the manuscript. There are also some minor changes requested to the structure and wording of the manuscript. Details are listed in the specific comments section.

Specific comments

Line 60 –Did authors mean that AK may lead to the loss of the eye or loss of the eye functionality or simply to loss of vision? Please reconcile to be more specific.

Line 84, 89, 121, 205, 217 – please italicize “A. castellanii”

Line 94, 95 - please italicize  “E. aerogenes”

Line 237  - please italicize  “in vivo”

Line 313  - please italicize  “in vitro”

Results:

Section 2.1. Amoebic Cultivation should be incorporated into material and methods under already existing section 5.2 as it does not provide any original results but rather describes the methodology used.

Section 2.2. Reactivation of A. castellanii Virulence should be incorporated into material and methods under already existing section 5.3 – the same reason as above.

Conclusions:

In order to increase the scientific value of the work, authors should take into consideration and discuss the following interesting points that came out afer lecture of this manuscript:

  • Is there any correlation between level of trophozoites present round blood vessels and level of hematogenous dissemination to other organs in the tested mice?
  • Is there any correlation between route of infection (damaged skin (ulcers) or undamaged skin), and level of damages caused by the trophozoites on the histological level observed?

Round 2
